

# Genome-wide identification of *ZmHMAs* and association of natural variation in *ZmHMA2* and *ZmHMA3* with leaf cadmium accumulation in maize

Yanhua Cao[1], Xiongwei Zhao[2], Yajuan Liu[1], Yalong Wang[1], Wenmei Wu[1], Yiwei Jiang[3], Changjian Liao[4], Xiaoxun Xu[5], Shibin Gao[1], Yaou Shen[1], Hai Lan[1], Chaoying Zou[1], Guangtang Pan[1] and Haijian Lin[1]

[1] Maize Research Institute, Sichuan Agricultural University, Chengdu, Sichuan, China
[2] College of Life Sciences, Shanxi Agricultural University, Jinzhong, Shanxi, China
[3] Department of Agronomy, Purdue University, West Lafayette, IN, USA
[4] Crop Research Institute, Fujian Academy of Agricultural Sciences, Fuzhou, Fujian, China
[5] College of Environment Sciences, Sichuan Agricultural University, Chengdu, Sichuan, China

Corresponding author
Haijian Lin, linhj521@gmail.com

## ABSTRACT

$P_{1B}$-type ATPases, known as heavy metal ATPases (HMAs), play an important role in the control of cadmium (Cd) accumulation in plants. In this study, a total of 12 *ZmHMA* genes were identified in the maize genome and particularly classified into six clusters based on their phylogenetic relationship and motif compositions. Furthermore, the expression patterns of different *ZmHMA* genes varied with developmental stages, and were tissue specific under normal conditions. *ZmHMA2* and *ZmHMA3* genes exhibited significant up-regulation under Cd treatment. Eventually, the association analysis between 103 inbred lines and alleles in *ZmHMA2* and *ZmHMA3* revealed that one insertion–deletion (InDel) in the intron from *ZmHMA2* was associated with leaf Cd concentration under low Cd condition at the seedling stage. Twenty polymorphisms in *ZmHMA3* were significantly associated with leaf Cd concentration under various Cd levels at seedling and maturing stages. Five single nucleotide polymorphisms (SNPs) and two InDels of these significantly associated polymorphic loci from *ZmHMA3* caused the amino acid substitutions and insertion or deletion events. Importantly, the proteins encoded by *ZmHMA2* and *ZmHMA3* genes were located in the plasma membrane. This comprehensive analysis will provide an important theoretical basis for future functional verification of *ZmHMA* genes to unravel the mechanisms of Cd accumulation in leaves of maize. Additionally, the favorable alleles in *ZmHMA3* will lay a foundation for the marker-assisted selection of low Cd accumulation in maize.

## INTRODUCTION

Heavy metal contamination in soil has become a severe problem in many parts of the world, which threatens human life through the food chain (*Li et al., 2014a*). Cadmium (Cd) is one of the most toxic heavy metals present in the soil environment and is readily

absorbed by vegetables and crop plants (*Huang, Ying & Yunxia , 2009*). Most of the Cd accumulation in the aboveground tissues, especially in the leaves of maize has become a main source of Cd intake in numerous developing countries as maize has been used as agricultural raw materials and as a feedstock for livestock for many years. Therefore, a better understanding of the mechanism of Cd accumulation in maize is an important step for breeding programs in developing varieties with low Cd accumulation.

$P_{1B}$-type ATPases, also named Heavy Metal ATPases (HMAs), belong to the P-type ATPase superfamily ($E_1$–$E_2$ ATPases) that transport various metal cations across biological membranes. Through utilizing the energy generated from ATP hydrolysis, some HMA transporters act as an efflux pump to load heavy metals into the xylem from the surrounding tissues. In *Arabidopsis*, AtHMA2 and AtHMA4 are localized in the plasma membrane (PM) and are responsible for Cd and Zn efflux from cells (*Jin et al., 2015*; *Verret et al., 2004*). Overexpression of *AtHMA4* enhances root-to-shoot translocation of Cd. In rice, OsHMA2 is also localized in the PM, and plays a role in Zn and Cd loading to the xylem and participating in root-to-shoot translocation (*Satoh-Nagasawa et al., 2011*; *Takahashi et al., 2012b*). Overexpression of *OsHMA2* significantly reduces Cd and Zn accumulation in the rice leaves compared to the wild type (*Takahashi et al., 2012b*). However, some HMA transporters appear to assist in metal uptake and homeostasis. AtHMA3 is localized in the vacuolar membrane and its overexpression induces accumulation of Cd and Zn in the shoots and roots of *Arabidopsis* (*Morel et al., 2009*). On the contrary, OsHMA3, also localized in the vacuolar membrane, limits Cd translocation from the roots to the aboveground tissues (*Ueno et al., 2010*). Given the diverse biological function of different *HMA* genes, their roles in Cd accumulation needs to be further explored in plant species.

Although some *HMA* genes have been functionally characterized in *Arabidopsis* and rice, the detailed characterization and functions of *HMA* family members for Cd translocation remain unknown in maize. In the present study, we investigated phylogenetic relationship, gene structure, expression pattern, and motif composition of *HMA* family genes in maize. In addition, association analysis between the genetic variations of *ZmHMA2* and *ZmHMA3* and plant response to Cd stress was conducted in a diverse maize population consisting of 103 inbred lines from tropical and temperate regions. The objective of the study was to identify specific *ZmHMA* genes that control Cd transport and to analyze the loci causing functional variation of the target genes. In addition, the favorable alleles derived from polymorphic loci within *ZmHMA2* and *ZmHMA3* will provide the foundation for the marker-assisted selection of low Cd accumulation in maize breeding.

## MATERIALS AND METHODS

### Plant Materials and Phenotyping

Two experiments were conducted for analyzing Cd accumulation in leaves of 269 inbred lines. Detailed information of experimental design and plant growing conditions were described previously (*Zhao et al., 2018*). Briefly, for a controlled environment study, maize seedlings of 269 inbred lines were planted in plastic pots containing 14 kg clayey soil under natural sunlight in a greenhouse in 2015 and 2016, respectively. In order to simulate the

extent of Cd pollution of urban soil in China (*Li et al., 2014b*), the middle-Cd treatment were applied as $CdCl_2 \cdot 2.5H_2O$ at 0.1 mmol $kg^{-1}$ (available Cd 18.8 mg $kg^{-1}$ in soil), while no $CdCl_2 \cdot 2.5H_2O$ was added to the soil for low-Cd treatment (available Cd of 3.28 mg $kg^{-1}$ in soil). Seedlings (18 d old) were cultured under low-Cd condition and middle-Cd conditions in a randomized complete block design with two replicates. After 15 days, the third and fourth leaves were harvested to analyze Cd accumulation in maize leaves. The traits for the pot experiment at seedling stage were labelled LCd15-Low (Leaf Cd concentration under low-Cd level in 2015), LCd16-Low (Leaf Cd concentration under low-Cd level in 2016), LCd15-Middle (Leaf Cd concentration under middle-Cd level in 2015), and LCd16-Middle (Leaf Cd concentration under middle-Cd level in 2016). For the field experiment, the inbred lines were planted in a highly Cd-contaminated soil (32.5 Cd mg $kg^{-1}$ soil) in Deyang city in 2015 and 2016. The field experiment was a randomized complete block design with two replications. After the seeds matured, five consecutive plants were chosen from the middle of each row, and the middle leaf below the tassel was harvested for measuring Cd concentration (LCd15-Field and LCd16-Field).

Harvested leaf samples were rinsed with tap water, washed three times with deionized water, and then dried at 70 °C in an oven until completely dried. The dry samples were ground to a powder using a pestle with liquid nitrogen. Subsequently, the Cd concentration was determined by using inductively coupled plasma-atomic emission spectrometry (ICP-MS) as described previously.

## Identification of HMA family in maize and phylogenetic analysis

To identify the *HMA* genes in maize, the known amino acid sequences of the published cloned *OsHMA2* and *OsHMA3* (*Takahashi et al., 2012a*; *Ueno et al., 2010*) were used in the BLASTP program against the maize B73 reference genome (B73 RefGen_v3, https://www.maizegdb.org/). The protein was selected as a candidate protein only when passing the expected threshold of 1e−10, and amino acid sequence > 200 residues. Each annotated protein was examined for the existence of E1–E2 ATPase domain (PF00122) by SMART (http://smart.embl-heidelberg.de/). Multiple sequence alignment was performed by the ClustalW2.1 program with default parameters (*Larkin et al., 2007*). Transmembrane domains were predicted with internet-programs SOSUI (http://bp.nuap.nagoya-u.ac.jp/sosui/).

The phylogenetic analysis among maize, rice and *Arabidopsis* was conducted with the neighbor joining (NJ) method (1,000 bootstrap replicates) using ClustalW2.1 based on the full-length protein alignment. The phylogenetic tree was displayed in FigTree software. The exon-intron structure of *ZmHMA* genes was graphically displayed by the Gene Structure Display Server (*Hu et al., 2014*). The ZmHMA protein sequences were used to predict the conserved motifs by using the MEME Suite web server (http://meme-suite.org/) (*Bailey et al., 2009*) with the maximum number of motif sets at 10 and the optimum width of motifs from 5 to 300 amino acids.

## Gene expression analysis

Expression patterns of *ZmHMA* genes in different maize tissues were analyzed using the genome-wide gene expression atlas of maize inbred B73 line that was reported previously

(*Stelpflug et al., 2016*). Expression data under normal condition for the 6 tissues were combined from 79 distinct replicated samples (RNAseq) (Table S1). In addition, the responsiveness of each *ZmHMA* genes to Cd stress was analyzed by quantitative real-time PCR (qRT-PCR). Briefly, the 2-week-old plants of the B73 (a line with low Cd accumulation and a reference genome of maize) were grown in $\frac{1}{2}$ Hoagland's nutrient solution amended with $CdCl_2 \cdot 2.5H_2O$ ($200\,\mu mol\,L^{-1}$) for 0 h (used as control), 12 h, 24 h, and 48 h. Total RNA from the roots, stems and leaves was isolated using TRIZOL reagent (Invitrogen, USA) and then reverse transcription was performed with PrimeScript RT Reagent Kit (Takara, Japan). The primers used in the qRT-PCR experiments were designed by Primer 5.0 software and listed in Table S2. *GADPH* was used as a housekeeping gene for normalization in the present study. Subsequently, qRT-PCR was conducted using the SYBR premix Ex Taq kit (Takara) on an ABI 7500 Real-Time System (Applied Biosystems) as follows: 95 °C for 30 s; 95 °C for 5 s, 60 °C for 30 s, 40 amplification cycles, and then the melt curves were generated for verification of amplification specificity by a thermal denaturing step. The method of $2^{-\Delta\Delta CT}$ was used to calculate the relative gene expression level between times (*Schefe et al., 2006*). The analysis included three biological replicates and three technical replicates for each sampling time.

### *ZmHMA2* and *ZmHMA3* gene sequences and association analysis with cd accumulation in maize

In total, 103 maize inbred lines (stiff stalk, non-stiff stalk, and tropical or subtropical group) were randomly selected from a natural maize population of 269 inbred lines (Table S3), and used for an association analysis. To identify the sequence variations among candidate genes associated with Cd accumulation the *ZmHMA2* (*GRMZM2G099191*) and *ZmHMA3* (*GRMZM2G175576*), full sequences (5′-UTR, exons, intron and 3′-UTR) of these two genes were amplified from genomic DNA in 103 maize inbred lines using high-fidelity polymerase KOD FX Neo (TaKaRa). Three pairs of primers were designed using the B73 genome sequence as a reference and Primer 5.0 software (Table S2). A 150 bp overlap between each pair of primers was designed in the target regions. PCR products from 103 inbred lines were purified using E.Z.N.A and sequenced directly using the ABI 3730 sequencer. These sequences were aligned using MUSCLE 3.8, assembled using ContigExpress, and manually corrected using BioEdit 7.1 software (*Edgar, 2004*).

Nucleotide polymorphism in *ZmHMA2* and *ZmHMA3*, including SNP and InDels, was identified and extracted (MAF $\geq$ 0.05). Allelic diversities with parameters of nucleotide diversity ($\pi$) and nucleotide polymorphism ($\theta$) were calculated using DnaSP 6.0 software (*Rozas et al., 2017*). Tajima's D statistics and Fu and Li's statistical tests of *ZmHMA2* and *ZmHMA3* were selected to investigate the evolutionary pressure within the different regions. The significance of each DNA polymorphism associated with Cd leaf concentration of maize was calculated using a mixed linear model (MLM) implementing both population structure (Q) and kinship (K) in TASSEL 3.0 (*Bradbury et al., 2007*; *Liu et al., 2015*). The Q and K were calculated using genome wide SNPs as described by *Zhao et al. (2018)*. The significance threshold of $P < 0.01$ was used for the candidate gene-based association

analysis. Levels of linkage disequilibrium (LD) between pairs of two polymorphic loci were estimated using Haploview v4.2 (*Barrett et al., 2004*).

### Subcellular localization of ZmHMA2 and ZmHMA3 proteins

We used ProtComp version 9 (http://linux1.softberry.com/berry.phtml) for comparing homologous proteins of known localization and pentamer distributions in the LocDB and PotLocDB databases to predict the putative protein subcellular localization. For verifying the prediction, the coding regions of *ZmHMA2* and *ZmHMA3* without the stop codon were amplified from B73 cDNA and cloned downstream of the 35S promoter in the pCAMBIA2300 vector that carried GFP. The primers used to amplify *ZmHMA2* and *ZmHMA3* were listed in Table S2. The PCR products were digested with BamHI, and directionally ligated into vector pCAMBIA2300 to construct the ZmHMA2-GFP or ZmHMA3-GFP fusion gene driven by a 35S promoter (*Liu et al., 2010*), respectively. In addition, equal volume suspensions of *Agrobacterium* strain GV3101 harboring Ti plasmids expressing either *ZmHMA2-GFP* and a plasma marker (*PIP2A-RFP*) or *ZmHMA3-GFP* and a plasma marker (*PIP2A-RFP*) were mixed (final $OD600 = 0.6$). These plasmids were individually expressed or co-expressed in tobacco (*Nicotiana benthamiana*) leaves by *Agrobacterium*-mediated infiltration. GFP and RFP signals were visualized using a fluorescence confocal scanning microscope (Nikon A1 i90, LSCM, Japan) at 72 h after infiltration.

## RESULTS

### Identification and properties of putative maize *HMA* genes

$P_{1B}$-type ATPases (HMAs) in maize were identified using the cloned OsHMA2 (*Oryza sativa Japonica*) and OsHMA3 (*Oryza sativa Japonica*) as protein queries in BLASTP searches against the B73 reference genome sequence. A total of 12 *HMA* genes were identified and were named *ZmHMA1* to *ZmHMA12* (Table 1) following the nomenclature of rice. These *ZmHMA* genes were unevenly distributed along the 5 out of 10 maize chromosomes. Four *ZmHMAs* were present on chromosome 2; three on chromosome 5; and only one gene each on chromosomes 1 and 9. The relatively high densities of *ZmHMA* genes were observed at the central sections of chromosomes including the centromeres and the pericentromeric regions (Table 1). Only *ZmHMA6* and *ZmHMA11* were noted at the bottom of chromosome 4. Gene structure analysis revealed that most *ZmHMA* genes displayed the complex exon-intron structure. Seven *ZmHMA* genes have more than nine exons. The predicted molecular weights of the 12 deduced ZmHMA proteins ranged from 42.3 (ZmHMA12) to 112.43 kDa (ZmHMA2) (Table 1). These results indicated that the *HMA* genes were highly conserved over the course of evolution since the pericentromeric regions of chromosomes have reduced recombination rates.

### Phylogenetic and genomic structure analysis of maize HMA proteins

To investigate the phylogenetic relationship between the *HMA* gene family in maize, *Arabidopsis* and rice, the amino acid sequences of 12 HMA members from maize, 8 from *Arabidopsis* and 9 from *Oryza sativa* were used to construct a phylogenetic tree. Based on

Cao et al. (2019), *PeerJ*, DOI 10.7717/peerj.7877

**Table 1** List of annotated *HMA* genes in maize.

| Gene name | Primary transcript name | Chr | Bin | Genome location | Gene size (bp) | CDS size (bp) | Exons | Length (aa) | Molecular weight (kDa) | Theoretical isoelectric point (pI) |
|---|---|---|---|---|---|---|---|---|---|---|
| *ZmHMA1* | GRMZM2G067853_T02 | 5 | 5.03 | 52,146,625–52,168,252 | 21,627 | 2,472 | 13 | 823 | 87.86 | 6.67 |
| *ZmHMA2* | GRMZM2G099191_T01 | 5 | 5.03 | 56,451,481–56,458,107 | 6,626 | 3,300 | 10 | 1,099 | 117.86 | 6.30 |
| *ZmHMA3* | GRMZM2G175576_T02 | 2 | 2.06 | 159,040,587–159,044,553 | 3,966 | 2,697 | 5 | 898 | 92.56 | 6.67 |
| *ZmHMA4* | GRMZM2G455491_T01 | 2 | 2.06 | 159,019,693–159,023,011 | 3,318 | 2,895 | 5 | 964 | 96.02 | 7.84 |
| *ZmHMA5* | GRMZM2G144083_T01 | 2 | 2.04 | 19,244,753–19,250,983 | 6,230 | 2,556 | 6 | 851 | 112.43 | 6.34 |
| *ZmHMA6* | GRMZM2G315931_T01 | 4 | 4.08 | 197,229,916–197,239,315 | 9,399 | 2,787 | 17 | 928 | 97.36 | 7.10 |
| *ZmHMA7* | GRMZM2G029951_T01 | 5 | 5.04 | 121,445,070–121,449,579 | 4,509 | 2,943 | 10 | 980 | 106.28 | 5.33 |
| *ZmHMA8* | GRMZM5G855347_T01 | 1 | 1.03 | 16,331,867–16,338,054 | 6,187 | 2,640 | 16 | 879 | 101.48 | 6.61 |
| *ZmHMA9* | GRMZM2G404702_T01 | 9 | 9.04 | 109,908,973–109,914,816 | 5,843 | 3,072 | 10 | 1,023 | 109.52 | 5.31 |
| *ZmHMA10* | GRMZM2G143512_T01 | 2 | 2.04 | 19,329,257–19,337,335 | 8,078 | 3,000 | 6 | 999 | 102.37 | 6.46 |
| *ZmHMA11* | GRMZM2G010152_T01 | 4 | 4.10 | 236,277,989–236,283,316 | 5,327 | 2,997 | 9 | 998 | 107.58 | 5.33 |
| *ZmHMA12* | GRMZM2G151406_T01 | 9 | 9.04 | 109,906,735–109,908,497 | 1,761 | 1,200 | 4 | 400 | 42.3 | 5.6 |

the phylogenetic tree, it clearly showed that all the maize HMA proteins were classified into the same corresponding categories in *Arabidopsis* and *O. sativa*, which include HMA I, HMA II, HMA III, HMA IV, HMA V and HMA VI clusters (Fig. 1). Functional annotation against gene ontology terms (GO; http://www.geneontology.org) shown that all *ZmHMAs* were involved in ATPase-coupled transmembrane transporter activity, cation transport and ATP binding (Table S4). *ZmHMA1*, *ZmHMA1*, *ZmHMA3* and *ZmHMA4* (HMA I, HMA II) were involved in Cadmium/Zinc ion transmembrane transport (GO:0070574, GO:0071577) for biological process categories. Eight of these *ZmHMAs* (HMA III–HMA VI) were listed as involved in Copper (Cu) ion binding (GO:0005507) for molecular function. In addition, each cluster contained at least one *Arabidopsis* and rice member in the HMA clusters (Fig. 1, Fig. 2A), which suggests that these homologous genes may have derived from multiple duplications during the evolution.

Due to the different lengths and abundant diversity of the functional domains of the HMA proteins, domain features were used for diversity analyses. As shown in Fig. S1, a total of 10 conserved motifs were identified in ZmHMA proteins by the protein domain analysis, including motifs 10 and 8 encoding the heavy-metal-associated domain (HMA domain), motifs 3, 5, 2 encoding E1–E2 ATPase domain, and motifs 4, 1 encoding the hydrolase domain. As expected, all the ZmHMA proteins possess the typical structures of the $P_{1B}$-type ATPases, including a E1–E2 ATPase domains (PF00122) (Fig. 2B). However, there were still obvious differences in the domain composition among these clusters. For instance, clusters V and VI had the largest number of motifs, while there only two motifs in cluster I. The HMA domain was absent in ZmHMA1 and ZmHMA12. ZmHMA1, ZmHMA2 and ZmHMA8 lacked the hydrolase domain (Fig. 2B). Analysis of transmembrane helices suggested that the proteins of cluster I had 6 to 9 predicted transmembrane domains (TMs) (Fig. S2, Table S5). Notably, *ZmHMA3* and *ZmHMA4* had exact similarity of gene structure on the same chromosome (Fig. 2C).

## Expression analyses of *ZmHMA* genes

To investigate the expression patterns of *ZmHMA* genes in different tissues, an expression heatmap was constructed for 12 *ZmHMA* genes in the 13 different tissues of the B73 under normal conditions (Fig. 3A). The results indicated that the expression patterns of different *ZmHMA* genes varied greatly among tissues. Of them, the transcripts of *ZmHMA2*, *ZmHMA6*, *ZmHMA7*, *ZmHMA11* and *ZmHMA12* were highly and constitutively expressed in the various tissues. The *ZmHMA3* transcript was expressed at a relatively high level in roots and nodes compared to other tissues. On the contrary, *ZmHMA1* and *ZmHMA4* were not expressed or exhibited at extremely low level in different tissues (Fig. 3A).

To determine which *ZmHMA* genes responded to Cd stress, the expression level of 12 *ZmHMA* genes was analyzed in B73 roots, stems, leaves under the Cd treatment using qRT-PCR. *ZmHMA4* failed to be amplified in roots, leaves and stems. We found that the expression levels of *ZmHMA1*, *ZmHMA3*, *ZmHMA5* and *ZmHMA12* in the roots were significantly upregulated by the Cd treatment (Fig. 3B, Fig. 3D, Fig. 3E, Fig. 3L). Among these genes, *ZmHMA3* was highly induced in roots and was upregulated more than 21-fold after 48 h of Cd treatment compared to normal growing conditions. In

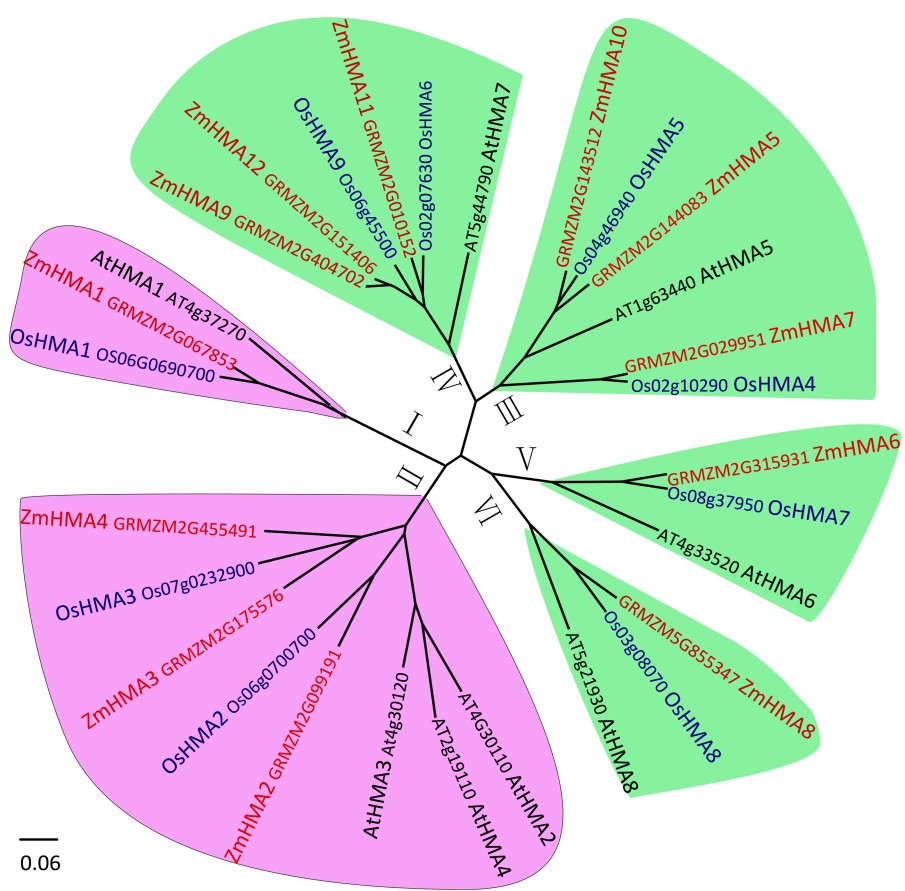

**Figure 1** **Phylogenetic analysis of HMA proteins between maize (ZmHMA1 to ZmHMA12), rice (OsHMA1 to OsHMA9) and *Arabidopsis* (AtHMA1 to AtHMA8).** On the basis of genetic and functional studies of the P$_{1B}$-ATPases, zinc (Zn)/cobalt (Co)/cadmium (Cd)/lead (Pb) group was highlighted in purple, the copper (Cu)/silver (Ag) group was highlighted in green. The number from I to VI represent different six branches.

leaves, the expression levels of *ZmHMA2*, *ZmHMA3*, *ZmHMA7* and *ZmHMA12* were significantly induced by Cd treatment (Fig. 3B, Fig. 3C, Fig. 3G, Fig. 3L). Specifically, the remarkably higher expression of *ZmHMA2* was found in leaves with a 15-fold increase after 12 h of Cd treatment. The other nine *ZmHMA* genes, *ZmHMA1*, *ZmHMA2*, *ZmHMA5*, *ZmHMA6*, *ZmHMA7*, *ZmHMA8*, *ZmHMA9*, *ZmHMA10* and *ZmHMA11*, were markedly downregulated in the stems. Collectively, the data indicated that different *ZmHMA* genes exhibited variable levels of expression in different tissues and developmental stages of maize.

### *ZmHMA2* and *ZmHMA3* gene polymorphisms and LD decay

Previous functional research reported that *OsHMA1-OsHMA3* (*Oryza sativa Japonica*) belonged to zinc (Zn)/cobalt (Co)/cadmium (Cd)/lead (Pb) transport (*Takahashi et al., 2012a*). OsHMA2 plays an important role in root-to shoot translocation of Zn and Cd, and OsHMA3 is important for limiting the Cd in above-ground tissues (*Takahashi et al., 2012b*;

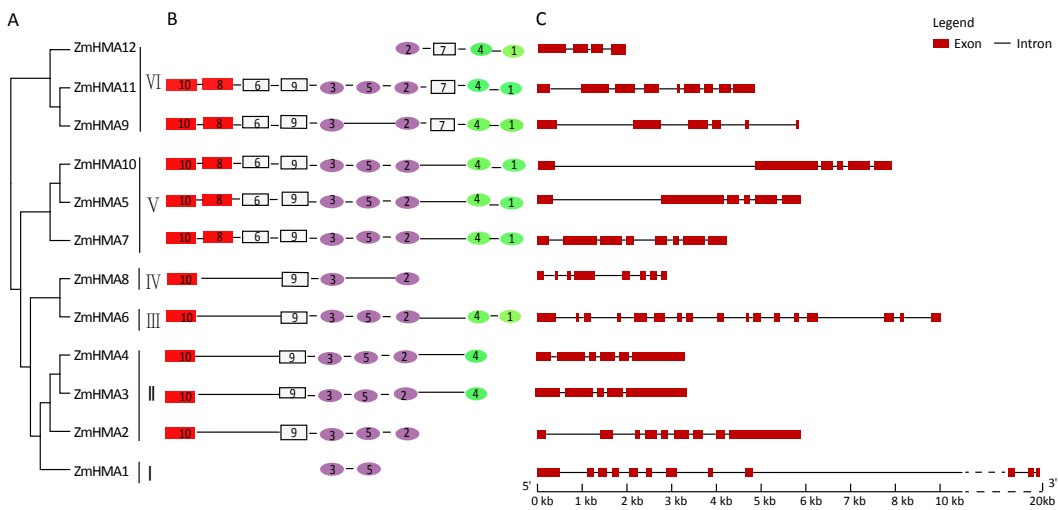

**Figure 2  Phylogenetic relationships (A), motif compositions (B) and gene structure (C) within the HMA family in maize.** Different motifs were numbered from 1 to 10 and motifs labeled in the same color indicate that the same conserved domains were present in the different motifs.

*Ueno et al., 2010*). On the basis of phylogenetic analysis, the ZmHMA2 protein exhibited 70% identity with OsHMA2 and ZmHMA3, and 72% with OsHMA3. Comprehensive analysis highlighted *ZmHMA2* and *ZmHMA3*, as candidate genes of particular interest.

To determine whether *ZmHMA2* and *ZmHMA3* genes were associated with Cd accumulation in leaves of maize, the natural variations within *ZmHMA2* and *ZmHMA3* genes were detected by determining the levels of nucleotide diversity. Sequence polymorphisms were detected among 103 maize inbred lines across 6,626 bp of *ZmHMA2* and 3,966 bp of *ZmHMA3*, which covered 5′-UTR, exons, introns and 3′-UTR. SNPs and InDels at the *ZmHMA2* and *ZmHMA3* locus were identified (Tables S6, S7). From the putative genomic sequences in the 103 inbred lines, 205 SNPs and 62 InDels were detected from *ZmHMA2*, with one SNP and InDel every 23 and 103 bp, respectively (MAF $\geq 0.05$) (Table S6). In addition, 139 SNPs and 24 InDels events were found in *ZmHMA3*, with one SNP and one InDel every 27 and 157 bp, respectively (Table S7). Nucleotide diversity analysis using sliding windows indicated that introns of *ZmHMA2* had the highest nucleotide diversity ($\pi = 17.58 \times 10^{-3}$) and its exon had the lowest nucleotide diversity ($\pi = 6.24 \times 10^{-3}$) in maize (Table 2). However, in *ZmHMA3*, the estimates of nucleotide diversity in the 3′-UTR ($\pi = 33.57 \times 10^{-3}$) was obviously higher than that of other regions. Fu and Li's *F* and *D* test of each region showed that the exon of *ZmHMA2* and the 3′ UTR of *ZmHMA3* were under purifying selection with significantly positive values. These results suggested that purifying selection and/or population size expansion had occurred in the two regions respectively. In addition, the combined LD analysis of *ZmHMA2* and *ZmHMA3* showed LD decay was close to 0.1 within 1300 bp (Fig. 4).
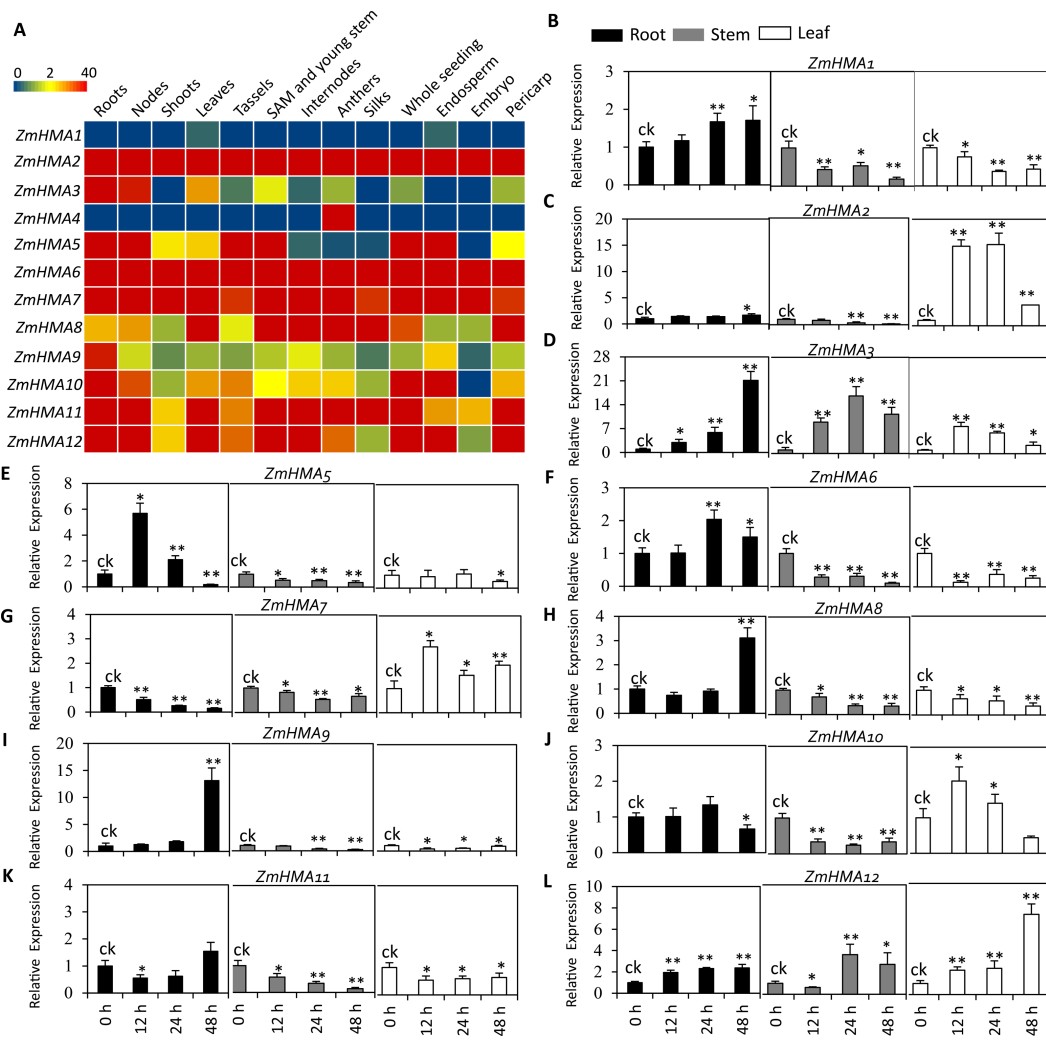

**Figure 3  Expression profiles of eleven *ZmHMA* genes in maize.** (A) A heat map illustrating levels of gene expression of the 12 *ZmHMAs* in 13 different tissues from various developmental stages. FPKM values were shown in different colors representing expression levels as indicated by the scale bar. (B–L) Relative expression levels of 11 *ZmHMAs* in maize B73 root, leaf and stem tissues grown hydroponically under normal (ck) and Cd-additive conditions (200 μM) at the vegetative growth stage for 12 h, 24 h and 48 h. The comparisons of gene expression under ck and Cd stress conditions were performed using a two-sided t-test: * $p \leq 0.05$, ** $p \leq 0.01$. Bars indicate standard deviation.

## *ZmHMA2* and *ZmHMA3* genes associated with Cd accumulation in leaves of maize

The association analysis between natural variations in *ZmHMA2* and *ZmHMA3* and Cd concentration in maize were performed using the Q+K model. One co-located InDel in the intron (InDel S1174, $P = 7.96 \times 10^{-3}$) in *ZmHMA2* was associated with leaf Cd concentration at seedling stage under low Cd condition in 2015 and 2016 (LCd15-Low and LCd16-Low) (Table 3). For *ZmHMA3*, 20 significant loci (15 SNPs and five InDels) were associated with leaf Cd concentration under various soil environments at seedling

**Table 2   Nucleotide diversity and neutrality test of *ZmHMA2* and *ZmHMA3*.**

| Gene | Regions | Length (bp) | InDel | SNP | Haplotype number | $\pi(\times 10^{-3})$ | $\theta\omega(\times 10^{-3})$ | Tajima's D | Fu and Li's D | Fu and Li's F |
|------|---------|-------------|-------|-----|------------------|-----------------------|--------------------------------|------------|---------------|---------------|
| | 5′-UTR | 322 | 4 | 9 | 6 | 9.38 | 1.54 | 2.01 | 1.26 | 1.73* |
| | CDS | 3,300 | 5 | 57 | 42 | 6.24 | 22.28 | −0.26 | −1.24 | −0.93 |
| *ZmHMA2* | Intron | 2,600 | 50 | 128 | 75 | 17.58 | 43.02 | 0.09 | −0.56 | −0.31 |
| | 3′-UTR | 409 | 3 | 11 | 12 | 10.82 | 4.23 | 0.12 | −1.15 | −0.77 |
| | Overall | 6,631 | 62 | 205 | 85 | 11.05 | 71.06 | 0.04 | −0.81 | −0.48 |
| | 5′-UTR | 187 | 0 | 3 | 7 | 6.26 | 1.34 | −0.30 | 0.31 | 0.12 |
| | CDS | 2,923 | 5 | 68 | 39 | 7.36 | 21.51 | −0.12 | −1.48 | −1.02 |
| *ZmHMA3* | Intron | 413 | 9 | 33 | 25 | 33.57 | 8.83 | 1.40 | 0.13 | 0.76 |
| | 3′-UTR | 447 | 10 | 35 | 17 | 18.85 | 8.83 | −0.29 | 1.55* | 0.90 |
| | Overall | 3,970 | 24 | 139 | 48 | 11.20 | 40.52 | 0.18 | −0.39 | −0.16 |

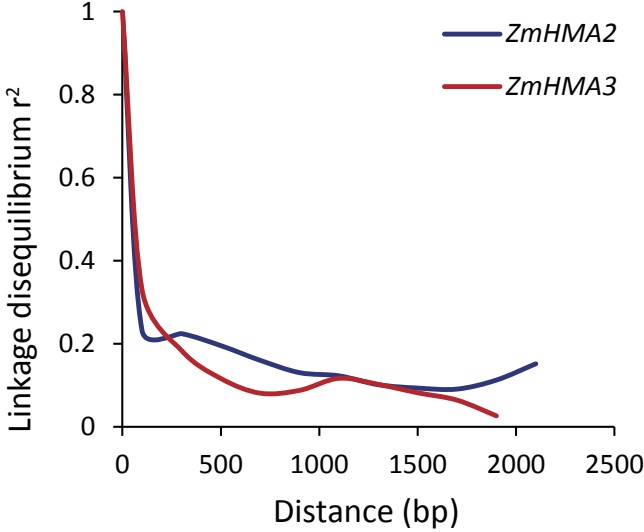

**Figure 4   Plot of linkage disequilibrium ($r^2$) decay against physical distance between SNPs from *ZmHMA2* and *ZmHMA3*, respectively.** Measurement was performed by averaging $r^2$ values of allele frequencies over a distance of 100 bp and plotting the values against distance (bp).

and maturing stages in 2015 and 2016 ($P < 0.01$) (Fig. 5, Table 3). $R^2$ marker values ranged from 7.7 to 16.3% among all those associated loci. On average, each site explained 11.2% (±3.3%) of phenotypic variance for its associated trait. Among the 20 loci, S917 located on the second exon region were significantly associated with LCd16-Middle ($P < 0.01$) (Table 3). Moreover, eight loci (S3097, S3114, S3297, InDel S3517, InDel S3594, S3692, S3695, S3792) with a complete LD were significantly associated with LCd15-Field and LCd16-Field. Furthermore, five SNPs and two InDels were located on the CDS region and caused the amino acid substitutions and insertions or deletions.
**Table 3** Associations between the natural variations within *ZmHMA2* and *ZmHMA3* genes and leaf Cd concentration under various Cd conditions by using kinship (K) and a population structure controlled model.

| Gene | Trait | Site | Allele[a] | MAF | $P(\times 10^{-3})$ | $R^2$ (%) | Amino acid variations |
|---|---|---|---|---|---|---|---|
| *ZmHMA2* | LCd15-Low | 1,174 | 3/0 | 0.24 | 7.96 | 7.57 | |
| | LCd16-Low | | | | 8.99 | 8.15 | |
| | LCd16-Field | 720 | C/A | 0.21 | 6.60 | 8.35 | |
| | LCd15-Middle | | | | 8.47 | 7.66 | |
| | LCd16-Middle | 917 | C/G | 0.18 | 2.09 | 10.47 | ALA (A)/GLY (G) |
| | LCd16-Field | 921 | C/G | 0.21 | 6.60 | 8.35 | |
| | LCd15-Middle | | | | 8.47 | 7.66 | |
| | LCd15-Low | 942 | C/G | 0.47 | 3.32 | 10.64 | |
| | LCd16-Field | 944 | C/G | 0.27 | 2.48 | 10.47 | ALA (A)/GLY (G) |
| | LCd15-Low | 951 | A/C | 0.46 | 3.32 | 10.64 | |
| | LCd16-Middle | 957 | G/A | 0.17 | 2.85 | 9.81 | |
| | LCd15-Low | 1,467 | C /T | 0.70 | 5.48 | 9.46 | |
| | LCd15-Low | 1,522 | T/A | 0.32 | 4.04 | 10.17 | |
| | LCd15-Low | 1,526 | 2/0 | 0.69 | 4.04 | 10.17 | |
| | LCd16-Field | 2,540 | T/A | 0.09 | 0.48 | 14.17 | SER (S)/THR (T) |
| | LCd16-Field | 2,809 | C/G | 0.12 | 0.99 | 12.52 | |
| *ZmHMA3* | LCd15-Field | 3,097 | G/A | 0.09 | 5.44 | 8.33 | |
| | LCd16-Field | | | | 0.21 | 16.15 | |
| | LCd15-Field | 3,114 | G/C | 0.09 | 5.44 | 8.33 | GLY (G)/ALA (A) |
| | LCd16-Field | | | | 0.21 | 16.15 | |
| | LCd15-Field | 3,297 | T/C | 0.09 | 5.44 | 8.33 | VLA (V)/ALA (A) |
| | LCd16-Field | | | | 0.21 | 16.15 | |
| | LCd15-Field | 3,517 | 9/0 | 0.09 | 5.44 | 8.33 | VLA (V)/Pro (P)/Stop |
| | LCd16-Field | | | | 0.21 | 16.15 | |
| | LCd15-Field | 3,594 | 6/0 | 0.09 | 5.44 | 8.33 | ALA (A)/Leu (L) |
| | LCd16-Field | | | | 0.21 | 16.15 | |
| | LCd15-Field | 3,692 | C/A | 0.09 | 4.97 | 8.76 | |
| | LCd16-Field | | | | 0.21 | 16.28 | |
| | LCd15-Field | 3,695 | 0/9 | 0.09 | 5.44 | 8.33 | |
| | LCd16-Field | | | | 0.21 | 16.15 | |
| | LCd15-Field | 3792 | 0/2 | 0.09 | 5.44 | 8.33 | |
| | LCd16-Field | | | | 0.21 | 16.15 | |

Notes.

[a]The letters indicate nucleotide polymorphisms and the numbers indicate the inserted or deleted nucleotides. Under lined letters and numbers represent the minor alleles. MAF represent minor allele frequency. $R^2$ represent explained phenotype variation.

## Putative *ZmHMA2* and *ZmHMA3* effect on Cd accumulation

Based on the average phenotype of plants grown in 2015 and 2016, the effects of gene polymorphisms on Cd accumulation were further analyzed using mixed linear model. Genotype with a deletion of GCA (InDel S1174) in an intron of *ZmHMA2* exhibited a significantly higher Cd concentration than that of the genotype with an insertion of GCA (InDel S1174) under low Cd condition at seedling stage (Fig. 6A). The individuals carrying the minor frequency alleles had a 1.03 mg lower leaf Cd content than those with

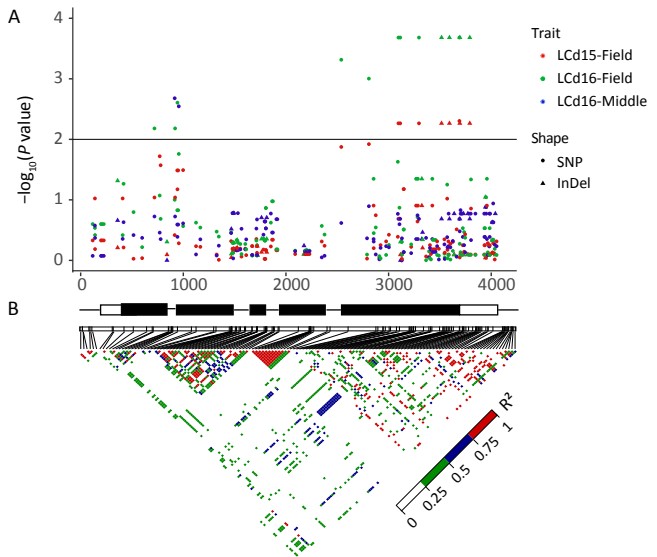

**Figure 5** **Natural variation within _ZmHMA3_ was associated with Cd concentration in leaves.** (A) Natural variation in _ZmHMA3_ associated with Cd concentration in leaves under different Cd conditions at seeding stage and mature stage. The $P$ value is shown on a $-\log_{10}$ scale. SNP represent single nucleotide polymorphism. InDel represent insertion or deletion. A schematic diagram of the entire gene structure is presented as the $x$-axis, including white and black boxes showing as UTRs and exons, respectively. (B) The pattern of pair wise linkage disequilibrium (LD) of DNA polymorphisms (MAF > 0.05) in decay of _ZmHMA3_. The level of LD ($r^2$) values is indicated with color key.

major frequency alleles ($P = 8.48 \times 10^{-3}$). In addition, the site S917 (CC) associated with LCd16-Middle at seedling stage had one amino acid substitution from alanine (A) to glycine (G) ($P < 2.09 \times 10^{-3}$). The Cd concentration of leaves in the genotype at seedling stage with SNP G/G was about 6.9 mg higher than those with SNP CC (Fig. 6B). In the field study across 2015 and 2016, four polymorphic loci (S3114, S3297, InDel S3517, InDel S3594) in the CDS region of _ZmHMA3_ were significantly co-associated with LCd16-Field and LCd15-Field. From the two SNP positions, two amino acid substitutions, from glycine (G) to alanine (A) at residue 3114 and from valine (V) to alanine (A) at residue 3296, were identified (Table 3). More importantly, InDel S3517 (GTCCCGTGA) encoded valine (V), proline (P) and the stop codon. In addition, InDel S3517 and the three loci above showed a complete LD pattern. A significant difference in Cd accumulation of leaves among the accessions with homozygous allelic SNP S3114, S3297 and InDel S3517 in _ZmHMA3_ was found at mature stage in the field study (Fig. 6C). The inbred lines with major frequency alleles (GG +TT+9) exhibited a significantly lower Cd accumulation in leaves than that of inbred lines with minor frequency alleles (CC+CC+0) ($P < 2.82 \times 10^{-3}$). The Cd accumulation in mature leaves of individuals with minor frequency alleles was significantly reduced by 10.96 mg compared to those with major frequency alleles.

## Subcellular Localization of ZmHMA2 and ZmHMA3 Proteins
Possible subcellular locations of ZmHMA2 and ZmHMA3 proteins were predicted using ProtComp 9.0. This prediction suggested that ZmHMA2 and ZmHMA3 might be localized
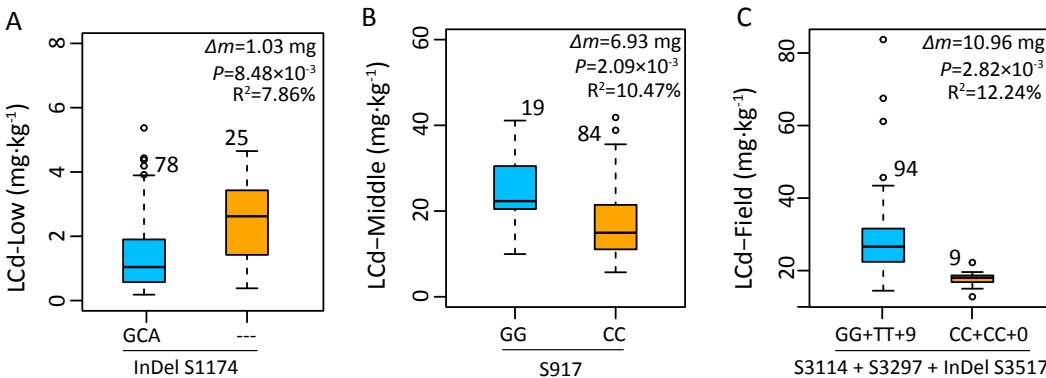

**Figure 6 Genetic effects of Cd accumulation in leaves were determined by significant SNPs or InDels in *ZmHMA2* and *ZmHMA3*.** The number represents the number of inbred lines homozygous for a determined allelic variant. $\Delta m$, the difference in mean leaf Cd concentration between minor alleles and major alleles over two years and two replications. Differences between the alleles were analyzed by mixed linear model (MLM, Phenotype SNP + Structure + Kinship). $R^2$ values from MLM of the data show the percentage the effect of the allele. (A) Allelic variations of InDel S1174 (3/0) in *ZmHMA2* associated with leaf Cd concentration under low Cd condition at seeding stage. (B) Allelic variations of S917 (G/C) in *ZmHMA3* associated with leaf Cd concentration under high Cd condition at seeding stage. (C) Multi-allelic variations of S3114 (G/C), S3297 (T/C), InDel S3517 (9/0) in *ZmHMA3* associated with leaf Cd concentration in the field.

on PM (Table S8). To examine the predictions above, the control vector *pCAMBIA2300-35S::GFP*, the loaded vector *pCAMBIA2300-35S::ZmHMA2-GFP* and *pCAMBIA2300-35S::ZmHMA3-GFP*, were transferred into tobacco using transient transformation. ZmHMA2 and ZmHMA3 proteins were successfully expressed as fluorescent protein fusions (ZmHMA2-GFP and ZmHMA3-GFP) (Fig. 7). Compared with the control (Fig. 7D), the green fluorescence signals of *ZmHMA2-GFP* (Fig. 7E) and *ZmHMA3-GFP* (Fig. 7F) were preferentially detected in the cytoplasm of the tobacco leaf epidermis. To further confirm subcellular localization of proteins, we co-infiltrated the tobacco leaves with *Agrobacterium* containing the PM marker (*PIP2A-RFP*) with *ZmHMA2-GFP* (Fig. 7H) and *ZmHMA3-GFP* (Fig. 7I), respectively. Confocal microscopy analysis of co-infiltrated cells indicated that GFP green signals of *ZmHMA2-GFP* (Fig. 7K) and *ZmHMA3-GFP* (Fig. 7L) overlapped with the RFP signals of the PM marker. These results demonstrated that ZmHMA3 and ZmHMA3 proteins were primarily localized on the PM.

## DISCUSSION

The $P_{1B}$-type ATPases, known as HMAs, play an important role in metal transport in plants. In recent studies, many *HMA* genes have been identified and researched in *Arabidopsis* and rice (*Jin et al., 2015*; *Takahashi et al., 2012a*). However, a thorough analysis of *HMA* gene family in maize are still relatively few. In this study, we conducted a comprehensive analysis of the *ZmHMA* gene family, including identification of members, phylogenetic relationships, expression profiles in different tissues and under Cd stress conditions, and candidate gene associations with leaf Cd concentration. The results will assist in

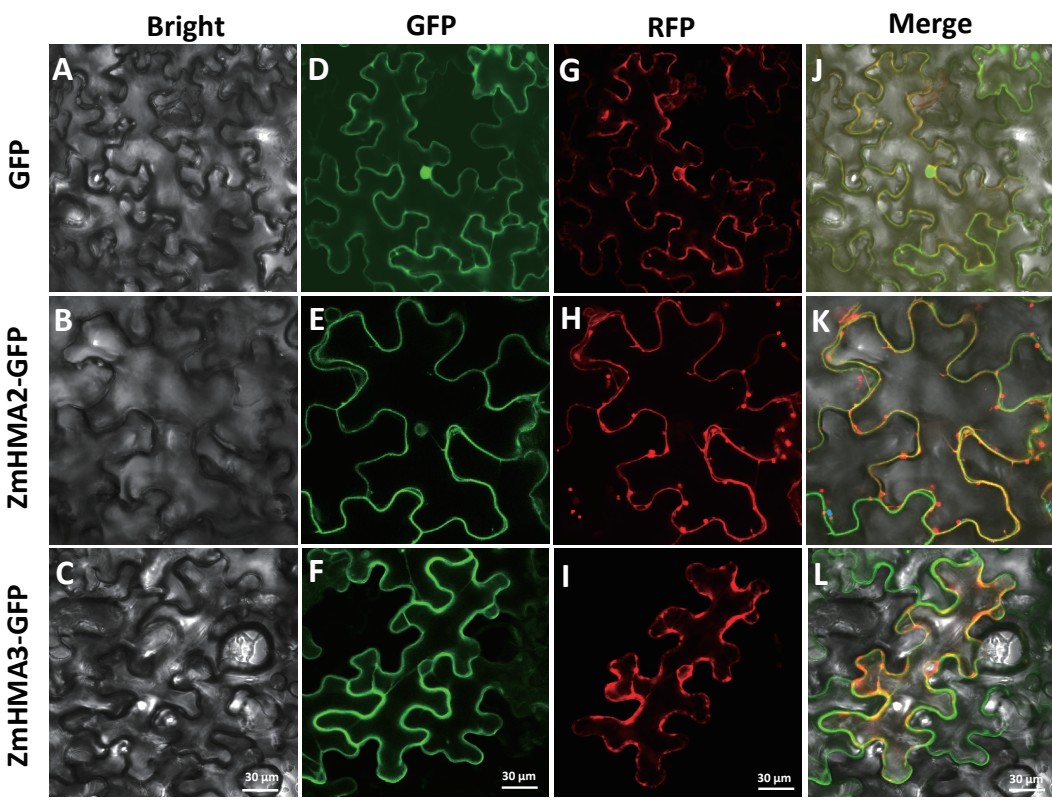

|  | Bright | GFP | RFP | Merge |

**Figure 7  Subcellular localization of ZmHMA2 and ZmHMA3 in leaves.** (A–C) Tobacco cells under bright-field illumination. (D–F) Location of the fusion proteins, including the empty vector control, *ZmHMA2 -GFP* and *ZmHMA3 -GFP* under GFP-field. (G–I) show co-expression of a plasma marker (PIP2A-RFP), *ZmHMA3 -GFP* and *ZmHMA3 -GFP* under RFP-field. (J–L) Merged bright-field, GFP-field and RFP-field images. Scale Bars, 30 mm.

understanding the roles of these *ZmHMA2* and *ZmHMA3* genes and their potential molecular mechanisms in response to Cd stress.

A total of 12 *ZmHMA* genes were identified in the maize genome using sequence comparison and phylogenetic analysis, the number of which was similar in *Arabidopsis* (*AtHMA1* to *AtHMA8*) and rice (*OsHMA1-OsHMA9*) (*Jin et al., 2015*) (Fig. 1). The gene structure analysis of ZmHMA proteins indicated that the *HMA* genes were highly conserved over the course of evolution since the pericentromeric regions of chromosomes have reduced recombination rates. In addition, most of HMA family members have one or more highly conserved N-terminal and/or C-terminal metal binding sits (MBDs) (*Williams & Mills, 2005*). Although *ZmHMA1-ZmHMA12* shared the E1–E2 ATPase domain, the diversity of motif compositions was still large with the number ranging from 2 to 10 motifs. Common motifs 8 and 10 that were characteristic of heavy-metal-associated domain were detected in the N-terminus and were included in the transmembrane (Fig. 3B). These variations in motif compositions indicated that specific conserved motifs in the HMA proteins underwent functional divergence to transport different metals. Plant HMAs have a diverse C-terminal region, such as His-rich (HXH, HH) and Cys-Cys dipeptides (CC,

CXC) (*Eren et al., 2006*; *Williams & Mills, 2005*). The C-terminal domain play regulatory roles, such as acting as a metal sensor to regulate the activity of the pump or to interact with pump regulatory proteins (*Mills et al., 2010*). In the present study, ZmHMA2 protein has a long C-terminal extension of 392 amino acids with 5 cysteine pairs and 16 histidine repeats (HH, HXH) towards the C-terminus. ZmHMA3 protein has a much shorter C-terminal region (156 amino acids), with 6 cysteine pairs and no histidine stretch (Fig. S1). The different types of MBDs and their positions in the plant $P_{1B}$-ATPases indicated ZmHMAs had various protein functions for metal mobilization and transportation.

Analysis of the temporal and spatial expression patterns of *ZmHMA* genes may provide useful information for establishing their putative functions. Therefore, we identified tissue-specific expression of *ZmHMA* genes under a normal growth condition. Our expression result showed that *ZmHMA2*, *ZmHMA6*, *ZmHMA7*, *ZmHMA11* and *ZmHMA12* were highly and constitutively expressed in the developing tissues, suggesting that these genes should be involved in maintaining the essential metal element during the developmental process (Fig. 3A). For instance, OsHMA9 were closely related to ZmHMA11 and function in Cu mobilization from mature leaves to young leaves (*Lee et al., 2007*). However, only three genes (*ZmHMA3*, *ZmHMA4* and *ZmHMA9*) exhibited tissue-specific expression in roots, nodes or anthers.

After being induced by Cd, most of the *HMA* genes in shoots and leaves (*ZmHMA1*, *ZmHMA5*, *ZmHMA8*, *ZmHMA9*, *ZmHMA11*) were downregulated, whereas only the expression of *ZmHMA3* and *ZmHMA5* were upregulated (Fig. 3B). Based on genetic and functional studies of the $P_{1B}$-ATPases, these transporters were found in a wide range of organisms and were divided into two groups: copper (Cu)/ silver (Ag) and Zn/Co/Cd/Pb transporters (*Takahashi et al., 2012a*). For the Zn/Co/Cd/Pb subgroup in maize ZmHMA1-ZmHMA4), ZmHMA2 was more closely related to OsHMA2 in rice. OsHMA2 plays an important role in preferentially distributing Zn and Cd and participating in Zn and Cd transport to developing seeds in rice (*Takahashi et al., 2012b*). ZmHMA3 and ZmHMA4 are the homolog of OsHMA3 and may control the Cd translocation. As previously reported, OsHMA3 transports only Cd and limits Cd accumulation in the grains (*Ueno et al., 2010*). However, *ZmHMA1* and *ZmHMA4* were not expressed or exhibited at extremely low level in different tissues (Fig. 3A). Notably, the expression of *ZmHMA2* and *ZmHMA3* were clearly inducible in the both the shoots and leaves by Cd stress, indicating a possible role for these two candidate genes in regulating Cd transportation (Fig. 3B).

Supporting this premise is the fact that in the association analysis, the genetic variation of *ZmHMA2* and *ZmHMA3* was found to be linked to the Cd accumulation variation. After controlling for population structure and relative kinship, one InDel (InDel S1174) in the intron from *ZmHMA2* was significantly associated with LCd15-Low and LCd16-Low at seedling stage ($P < 0.01$, Table 3). This result suggested that *ZmHMA2* might play an important role in Cd accumulation under low Cd condition in maize seedlings. ZmHMA2 was more closely related to the OsHMA2 orthologue in rice (Fig. 1). Our findings were similar to that reported by *Yamaji et al. (2013)*, showing that OsHMA2 were involved in metal transport at the vegetative growth stage. There was no significant decrease in the leaf Cd concentration in *OsHMA2* mutated plants at the reproductive stage. OsHMA2 was

reported to be localized on the plasma membrane (*Takahashi et al., 2012b*). In the present study, the subcellular localization by transient expression in tobacco demonstrated that ZmHMA2 was also located on the plasma membrane (Fig. 7K). Our results suggested that ZmHMA2 had a similar function with OsHMA2 and plays a role in Cd translation at the vegetative growth stage.

In the *ZmHMA3*, 13 non-synonymous and 7 synonymous loci were significantly associated with leaf Cd concentration at seedling and at mature stages in 2016 (Table 3). Of these associated loci, 4 loci located in the CDS region and close C-terminus were significantly co-associated with LCd16-Field and LCd15-Field. Further genetic analysis revealed that the coding in the accession with InDel S3517 would be terminated in advance and the polymorphic loci had a greater effect on phenotypic variation for the field environment at the mature stage. The *ZmHMA3* seems to play an important role in the control of Cd accumulation at the maturity stage, not during the vegetative growth stage. Based on phylogenetic analysis, ZmHMA3 shows 72% identity with OsHMA3. Previous studies have shown that OsHMA3 is localized on the tonoplast of rice root cells to sequestrate Cd into the vacuoles in the roots, keeping Cd away from the above-ground tissues and limiting Cd accumulation in the grain (*Ueno et al., 2010*). In addition, a variant in C-terminal region perform regulatory roles in OsHMA3 activity (*Kumagai et al., 2014*). All these results indicated that the variants on the C-terminal region of ZmHMA3 played an important role in controlling Cd accumulation of maize during the entire growth period. In our study, the subcellular localization demonstrated that ZmHMA3 was located on plasma membrane, which was different from OsHMA3 which was located on the tonoplast. This difference might be explained by the special biological function of HMA3. However, these special biological processes and favorable polymorphisms loci need to be further examined in maize.

## CONCLUSIONS

The comprehensive analysis of *ZmHMA* genes would provide overall new insights into their potential involvement in heavy metal transport. In total, 12 members of *ZmHMA* gene family were identified in the maize genome, which were classified into six clusters according to the structural and functional properties. Identified *ZmHMA2* and *ZmHMA3* genes had a relatively high expression in leaf tissue and as candidate genes to control Cd accumulation in leaves of maize under various Cd conditions at seedling and mature stages. More importantly, a significant association in the *ZmHMA3* gene with leaf Cd concentration was detected in that the DNA polymorphisms in the gene CDS region and close C-terminus. The variants on the C-terminal region of ZmHMA3 may in turn contribute to the Cd transport development across the plasma membrane. Moreover, favorable markers were derived from polymorphic loci within *ZmHMA3* will represent a valuable genetic resource for limiting Cd accumulation in maize by marker-assisted breeding. Future studies can be performed for gene validation for the mechanism of Cd accumulation in maize.

### Funding

This work was supported by National Key Technologies Research and Development Program (No. 2018YFD0200707, 2018YFD0800600), Education Department of Sichan Province (No. 035Z2240), Public Welfare Industry Research of Fujian Province (No. 2017R1026-1). The funders had no role in study design, data collection and analysis, decision to publish, or preparation of the manuscript.

### Grant Disclosures

The following grant information was disclosed by the authors:
National Key Technologies Research and Development Program: 2018YFD0200707, 2018YFD0800600.
Education Department of Sichan Province: 035Z2240.
Public Welfare Industry Research of Fujian Province: 2017R1026-1.

### Competing Interests

The authors declare there are no competing interests.

### Author Contributions

- Yanhua Cao and Xiongwei Zhao conceived and designed the experiments, performed the experiments, analyzed the data, prepared figures and/or tables, authored or reviewed drafts of the paper, approved the final draft.
- Yajuan Liu performed the experiments, analyzed the data, prepared figures and/or tables, approved the final draft.
- Yalong Wang and Hai Lan performed the experiments, contributed reagents/materials/analysis tools, prepared figures and/or tables, approved the final draft.
- Wenmei Wu performed the experiments, prepared figures and/or tables, approved the final draft.
- Yiwei Jiang analyzed the data, prepared figures and/or tables, authored or reviewed drafts of the paper, approved the final draft.
- Changjian Liao and Yaou Shen analyzed the data, contributed reagents/materials/analysis tools, prepared figures and/or tables, approved the final draft.
- Xiaoxun Xu contributed reagents/materials/analysis tools, approved the final draft.
- Shibin Gao, Guangtang Pan and Haijian Lin conceived and designed the experiments, contributed reagents/materials/analysis tools, authored or reviewed drafts of the paper, approved the final draft.
- Chaoying Zou performed the experiments, approved the final draft.

### Data Availability

The raw data is available in Tables S1–S8, and Figs. S1 and S2.
## Supplemental Information

Supplemental information for this article can be found online at http://dx.doi.org/10.7717/peerj.7877#supplemental-information.

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
