# Peer review of "Genome-wide identification of ZmHMAs and association of natural variation in ZmHMA2 and ZmHMA3 with leaf cadmium accumulation in maize"

_PeerJ, doi:10.7717/peerj.7877_

## Round 0.1 · original submission · Major Revisions

Though one of the reviews is quite critical I think that the manuscript has sufficient substance for publication in PeerJ after Major revisions and clarifications of the reviewers' questions in a revised manuscript. Please explain all modifications of the manuscript in an accompanying letter to the editor.

Additional comments:
line 398: Change gens to genes
lines 263 and 266:Change seeding to seedling
lines 124 and 144: Check numbering of supplemental tables

Reviewer 1 ·

Basic reporting

The objective of the study was not clear whether to address food safety or Cd toxicity. Do they intended to develop marker-assisted selection from food safety perspective or identification of genes responsible for Cd toxicity?

Materials and Methods are not self explanatory. Results and discussions are poorly written. There are also very confusing words in both MM and result sections
For example :
line 123 what does it mean 79 growth stages ? There is no 79 growth stages in maize or other plants
line 124 Table S2 is not the correct document
219 : non limiting condition?
line 278: What does mean 20 sites ?

Experimental design

The study lacks basic concept of experimental design including replications, field experiment design and no clear evidence why they chose the soil Cd levels.
Examples :
1) Plant materials and phenotyping lines 74-95

Both field and greenhouse experiments do not have any biological replications. There is no clear experimental design for both field and greenhouse experiments, and do not include how they analyzed the field and greenhouse data.
No clear evidence why they chose the mentioned soil Cd concentration

2) Gene expression analysis lines 120-138

The experiment lacks biological replications. Furthermore, there is not clear experimental design, how they chose these time points and Cd concentration and analyzed the data. The experimental design was also poorly described or set up. It is not clear how many factors and levels of each factor were used.


3) DNA sequencing lines 139-148
Read depth or sequence coverage is not provided

4) lines 149-161 Gene based GWAS

In order to apply, MLM model with accounting the Q and K parameters, it is not clear how they estimated Q and K parameters.
Do they estimate the two parameters from the 279 SNP or from previous studies?
Do the number of SNPs sufficient enough for Gene based GWAS?

Validity of the findings

My major concern is that the data included in the article is not helpful to address the main aim of the article i.e. identify potential markers and develop marker-assisted selection. The candidate genes reported here are superficial, not explained more than 16% of the phenotypic variance. Furthermore, the experiments lack most the basic experimental design and analysis.

·

Basic reporting

Good

Experimental design

Good

Validity of the findings

Good

Additional comments

Line 28, In abstract, what is “seeding”?
Line 30, Please define “SNPs” in the abstract
Line 35, What does mean by “low Cd accumulation” in the abstract. Do you mean below permissible limit?
Line 397-398, please reframe the sentence
Line 402, please see above comment at line 35
I could not find phenotypic data like fresh mass, length of root and shoot, etc. of maize seedlings

---

## Round 0.2 · Minor Revisions

Thank you for addressing the reviewer comments. The revised manuscript is almost ready for publication.

One of our Section Editors has pointed out that 'there is a complexity of the data which would gain a beneficial utility if annotation terms can be attached to the sequences included within the manuscript. This is especially important since variations in developmental stages and tissue location associations were made. This information would provide great value in distinguishing the 12 ZmHMA genes from other related candidates, as these candidates would surely be distinguished from other ATPases identified.'

He also mentioned that 'journal manuscripts are often scanned by text-mining software that locates and extracts core data elements, like gene function. Adding standard ontology terms, such as the Gene Ontology (GO, geneontology.org) or others from the OBO foundry (obofoundry.org) can enhance the recognition of your contribution and description. This will also make human curation of literature easier and more accurate. None of this was visible. The ability to discern genes based on biological, cellular, and molecular attributes lead to enhanced classifications which can be connected to related works.'

I am therefore issuing a 'minor revisions' decision to allow you to add these annotations. It looks like modifications can easily be added to Supplemental Table S1 with rows for the three GO: classifications under each gene; however, mention of its availability should be included within the manuscript if not outright placed there.

In addition, please correct the typo: line 324: HAM to HMA

·

Basic reporting

Good

Experimental design

Good

Validity of the findings

Good

Additional comments
* * *

---

## Round 0.3 · accepted · Accept

Congratulations to acceptance of your manuscript.